# Carbapenem-Resistant *Klebsiella pneumoniae* Bacteremia: Counterbalance between the Endemic Load and the Infection Control Program in a Hospital

Amalia Papanikolopoulou [1,*], Louisa Vini [2], Athina Stoupis [3], Dimitra Kalimeri [4], Anastasia Pangalis [5], Genovefa Chronopoulou [5], Nikos Pantazis [6], Panagiotis Gargalianos-Kakolyris [3] and Maria Kantzanou [7]

1. Clinical Pharmacology Department, Athens Medical Center, 5–7 Distomou Str., 15125 Marousi, Greece
2. Radiotherapy Department, Athens Medical Center, 5–7 Distomou Str., 15125 Marousi, Greece; l.vini@iatriko.gr
3. Clinical Infectious Diseases Department, Athens Medical Center, 1 Delfon Str., 15125 Marousi, Greece; astoupi@yahoo.com (A.S.); pagargalianos@gmail.com (P.G.-K.)
4. Nurse Department, Athens Medical Center, 5–7 Distomou Str., 15125 Marousi, Greece; kalimeridim@yahoo.gr
5. Biopathology Department, Athens Medical Center, 5–7 Distomou Str., 15125 Marousi, Greece; a.pangalis@iatriko.gr (A.P.); g.chronopoulou@iatriko.gr (G.C.)
6. Department of Hygiene, Epidemiology and Medical Statistics, Faculty of Medicine, School of Health Sciences, National and Kapodistrian University of Athens, 75 Mikras Asias Str., 11527 Athens, Greece; npantaz@med.uoa.gr
7. Department of Microbiology, School of Medicine, National and Kapodistrian University of Athens, 75 Mikras Asias Str., 11527 Athens, Greece; mkantzan@med.uoa.gr
* Correspondence: amaliapapaniko@yahoo.com; Tel.: +21-68-62-206

**Abstract:** Carbapenem-resistant *Klebsiella pneumoniae* (CRKP) remains a significant public health threat, given the associated increased healthcare burden and mortality rate. The objective of the current study was to investigate the association between the incidence of CRKP bacteremia, antibiotic consumption, and infection control measures in a tertiary-care hospital spanning the years 2013–2018. The analyzed indices included the incidence of CRKP bacteremia, antibiotic consumption, the use of hand hygiene solutions, and isolation rates of multidrug-resistant (MDR) carriers. In the total hospital, the incidence of CRKP bacteremia exhibited an absolute decrease during the study period, although this decrease did not reach statistical significance. Antibiotics used to treat CRKP infections, including carbapenems, colistin, tigecycline, and fosfomycin, as well as all classes of antibiotics, correlated positively with an increased incidence of CRKP bacteremia. On the contrary, increased use of scrub disinfectant solutions correlated negatively with a decreased incidence of CRKP bacteremia (IRR: 0.74, 95%CI: 0.59–0.93, *p*-value: 0.008) in the Adults ICU. Additionally, increased isolation rates of MDR carrier patients correlated negatively with a decreased incidence of CRKP bacteremia (IRR: 0.35, 95%CI: 0.13–0.97, *p*-value: 0.044). In conclusion, the implementation of multimodal infection control measures in our hospital contributed to the containment of CRKP, particularly in specific hospital sectors. However, the study suggests the need for additional strategies to overcome the endemic plateau.

**Keywords:** carbapenem-resistant *Klebsiella pneumoniae* bacteremia; antibiotics; infection control interventions; hospital





## 1. Introduction

The pathogen *Klebsiella pneumoniae* (*K. pneumoniae*), is an opportunistic Gram-negative bacterium which typically colonizes human mucosal surfaces of the oropharynx and gastrointestinal tract [1]. When *K. pneumoniae* spreads to other tissues, a range of invasive and fatal infections have been described, such as pneumonia, urinary tract infection, bloodstream infection, meningitis, and wound or surgical site infection [2–5]. More susceptible seem to be patients with comorbidities, the elderly or immunodeficient patients [2,6].

However, with the recent appearance and dissemination of hypervirulent strains, healthy individuals have also become vulnerable to *K. pneumoniae* infection [7]. Furthermore, there has been a global increase since 2000, in the occurrence of infections due to *K. pneumoniae* isolates belonging to the carbapenem-resistant Enterobacteriaceae (CRE) group [8]. This is primarily attributed to the clonal and plasmid-mediated spread of CRKP strains producing carbapenemases, a kind of β-lactamase that can hydrolyze carbapenem antibiotics [9]. Bloodstream infections resulting from CRKP pose a significant public health threat, given their association with heightened mortality rates, direct healthcare burdens, and indirect socioeconomic losses [10,11].

From a nosocomial surveillance survey in Greece, a high mean incidence of CR pathogens was reported, with a crude 28-day mortality rate reaching 34.4% in acute-care hospitals, highlighting a substantial public health challenge [12]. Specifically, for CRKP strains, their prevalence escalated in 2002 due to the production of VIM carbapenemases, followed by the emergence of KPC, NDM, and OXA-48-like carbapenemases and subsequently becoming endemic in many healthcare facilities since 2013 and 2014 [13,14].

Over the past decade, international recommendations have been issued to emphasize infection prevention and control measures, implement active surveillance programs, and enhance antibiotic stewardship, all aimed at reducing the incidence of CRE, including CRKP [15–17]. Numerous studies have also explored interventions to improve antibiotic prescribing practices in hospital settings, particularly for carbapenems [18,19]. Additionally, there has been a focus on studies addressing nosocomial control interventions to reduce colonization and infection caused by CRKP [20,21]. These efforts collectively contribute to the global initiative to mitigate the impact of antimicrobial resistance and improve patient outcomes.

In our previously published work [22], we concentrated on the examination of infection control interventions and outcomes within a tertiary-care hospital featuring an active, education-based infection control program. Among the most notable interventions were the increased utilization of hand disinfectant solutions and the isolation of MDR carriers, indicating a commitment to active surveillance and contact precautions [23]. Furthermore, a pivotal outcome was the reduction in the consumption of advanced antibiotics, including carbapenems, signifying adherence to a restricted formulary. Overall, the impact of this six-year program on the incidence of total carbapenem-resistant Gram-negative bacteremia revealed a decreasing trend in the hospital, although statistically nonsignificant [22].

The present study focuses specifically on assessing the effectiveness of this infection control program in influencing the incidence of CRKP bacteremia within our hospital in Athens, Greece.

## 2. Materials and Methods

### 2.1. Study Design

The research was carried out prospectively at a 300-bed tertiary-care hospital in Athens, Greece, spanning from 1 January 2013 to 31 December 2018. This hospital comprises three sectors, as illustrated in Figure 1: an Adults Clinic housing Internal Medicine, Surgery, Hematology, and Oncology Departments, along with an intensive care unit (ICU); an Obstetrics and Gynecology Clinic with a neonatal ICU; and a Pediatrics Clinic with a pediatric ICU [22]. Throughout the study duration, there were no alterations in the types of ICUs, single-bed rooms, medical, nursing, and laboratory diagnostic procedures, all of which were overseen by the Quality Assurance Department. Furthermore, there were no changes in staffing policies.

## HOSPITAL SECTORS

**Figure 1.** Hospital Sectors; Dpts: Departments; ICU: Intensive care unit.

### 2.2. Interventions

Throughout the study period, the following interventions were implemented: 1. surveillance of CRKP, along with other MDR pathogens; 2. formulary restriction and preauthorization for advanced antibiotics; 3. promotion of hand hygiene, monitored in situ using the 3M™ Clean-Trace™ Hygiene Monitoring System, 4. active screening for MDR carriage through pharyngeal, axillary-rectal, and nasal cultures, and isolation of MDR carriers with isolation protocols in a single room both for colonized and infected patients with contact precautions; 5. standard environmental cleaning procedures and infection control measures.

### 2.3. Data Collection and Outcomes

Prospective data collection was conducted, and the following outcomes were assessed on a monthly basis: 1. incidence of CRKP bacteremia; 2. antibiotic consumption; 3. consumption of hand disinfectant solutions, including soap, scrub disinfectant solutions with chlorhexidine, and alcohol 70% disinfectant solutions with chlorhexidine [22].

### 2.4. Detection of Bacteremia and Microbial Resistance

Bacteremia was detected through Gram stains and blood cultures [22]. The automated VITEK 2 system (Biomerieux, Marcy-l' Etoile, France) was used for the isolation, identification, and antibiotic susceptibility testing. The CLSI breakpoints were used.

### 2.5. Definitions

Bacteremia was characterized as a laboratory-confirmed bloodstream infection, categorized as either primary (not associated with an infection at another body site) or secondary (believed to originate from a site-specific infection at another body site) [23]. A novel occurrence of bacteremia within a month, resulting from either a distinct pathogen strain or the same pathogen strain with a different resistance phenotype, was considered a new episode of bacteremia [24]. The total incidence of carbapenem-resistant Gram-negative bacteremia was determined by summing the incidences of CRKP, CR-*Pseudomonas aeruginosa*, and CR-*Acinetobacter baumanii* bacteremia, expressed per 1000 patient-days [22]. Data on antibiotic consumption were presented using the 2018 version of the World Health Organization Anatomical Therapeutic Chemical (ATC) Classification/defined daily doses (DDD) index, indicating the number of DDD per 100 patient-days. Advanced antibiotics, including carbapenems, colistin, tigecycline, fosfomycin, linezolid, daptomycin, ceftaroline, ceftazidime-avibactam, and ceftolozane-tazobactam, were specifically defined in order to restrict and preauthorize their prescription [22]. Hand hygiene consumption was defined as liters per 1000 patient-days. The isolation rate of MDR-carrier patients was expressed as the percentage of isolated patients per admissions [22]. Patients were screened for carriage if they were transferred from another hospital or with a history of recent hospitalization or coming from long-term care facilities.

*2.6. Statistical Analysis*

We examined temporal patterns in both intervention and outcome variables over the six-year study duration. The independent variable in the regression models was the time since January 2013 (commencement of the study), which was incorporated using appropriate restricted cubic splines. Fourier series terms of time (1st and 2nd order) were included in the models to account for potential seasonality effects. Standard errors (SE) and corresponding 95% confidence intervals (CI) were computed using the robust (sandwich) variance estimator to address potential violations of model assumptions [22]. Estimated values for the study period's beginning and end, along with corresponding 95% CIs, were derived by simplifying the models. Spline time terms were substituted with either a single linear time trend or two piecewise linear terms to capture the average long-term trend. Linear regression models were applied for antibiotic and disinfectant consumption. Poisson regression models were used for cases where bacteremia rate was the outcome of interest, employing the number of cases as the dependent variable and the appropriate number of patient-days as the offset after logarithmic transformation. Binomial regression models were utilized for cases where the outcome of interest was the percentage over the total number of hospitalizations (isolations), with the number of cases as the dependent variable and the appropriate number of hospitalizations as the binomial denominator [22]. We introduced relevant independent variables into the models to explore associations between outcomes and interventions. Initially, the effects of independent variables were tested separately for current ("month 0") and lagged values (months −1, −2, and −3). If the effects were statistically significant ($p$-value < 0.05) for more than one case (e.g., in month 0 and in month −1) and the association direction was consistent (e.g., positive for both), the average value was employed as the independent variable. In cases where the direction of the association differed (e.g., positive for "month 0" and negative for "month −1"), results of the respective models are presented separately. $p$-values were not adjusted for multiple testing [22]. The Stata version 14.2 (Stata Corp., College Station, TX, USA) was utilized for analyses.

### 3. Results

From January 2013 to December 2018, the hospital documented a total of 95,228 admissions. Concerning bacteremia, 1671 positive cultures were identified, constituting 7.58% of the 22,044 blood cultures conducted throughout the study period [22]. The calculated incidence of CRKP bacteremia stood at 0.09 per 1000 patient-days, while for total carbapenem-resistant Gram-negative bacteremia, it was 0.24 per 1000 patient-days. The incidences of CRKP in both the overall hospital clinics and the Adults ICU are visually presented in Figure 2.

Table 1 illustrates the time trends of CRKP bacteremia incidence rates. The incidence of CRKP bacteremia showed a decrease from 0.1 to 0.0 per 1000 patient-days in the total Hospital Clinics, while it increased from 0.3 to 0.5 per 1000 patient-days in the ICU. However, both changes were not statistically significant. When examining the percentage of relative change per year, there was a positive trend during the first two years of the study, although not statistically significant, in both the total Hospital Clinics and the Adults Clinic. This was followed by a statistically significant negative change over the last four years of the study (−27.10% [−42.97 to −6.81], $p$-value = 0.012 and −27.30% [−42.97 to −7.32], $p$-value = 0.010, respectively). In the Adults ICU, the percentage of relative change per year was statistically significant and positive during the first two years of the study (137.77% [8.59 to 420.63], $p$-value = 0.030) and marginally statistically significant and negative over the last four years of the study (−28.44% [−51.84 to 6.33], $p$-value = 0.098).

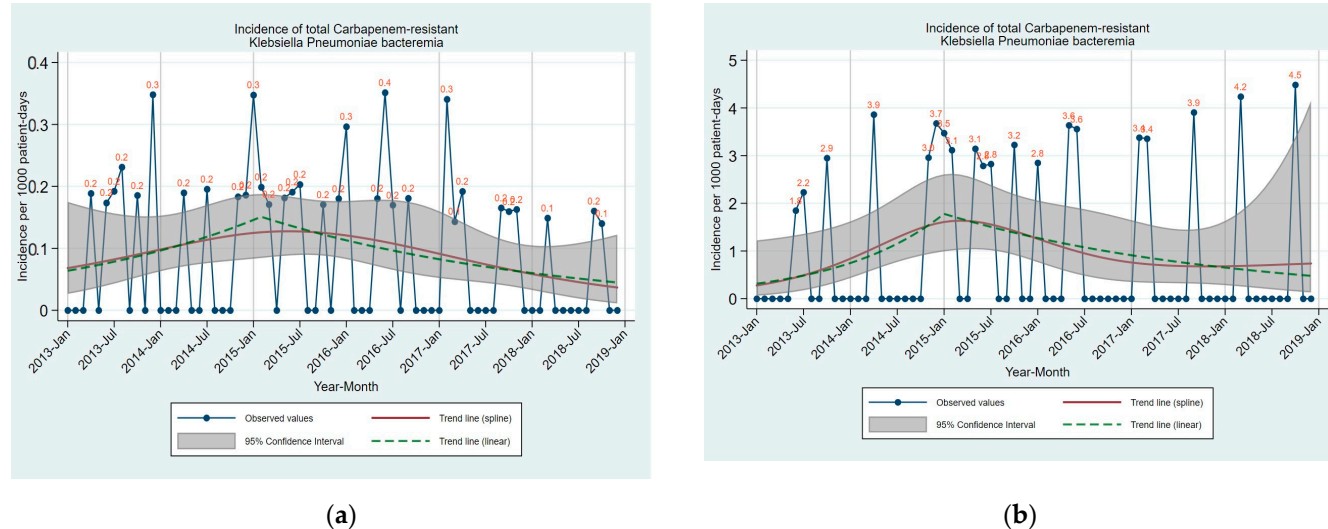

(**a**)                                                                                            (**b**)

**Figure 2.** Observed values and estimated time trends for total Carbapenem-resistant *Klebsiella pneumoniae* bacteremia/1000 patient-days, from January 2013 to December 2018: (**a**) in total hospital clinics, (**b**) in Adults ICU. ICU: intensive care unit; Jan: January; Jul: July.

**Table 1.** Time trend of bacteremia per 1000 patient-days in a hospital, January 2013 to December 2018.

| Incidence of Bacteremia /1000 Patient-Days | EVSP January 2013 (95% CI) | EVEP December 2018 (95% CI) | *p*-Value | %Relative Change/Year (95% CI) | *p*-Value |
|---|---|---|---|---|---|
| **Time Trend** | | | | | |
| **Total Hospital Clinics** | | | | | |
| Total CRKP Bacteremia | 0.1 (0.0 to 0.2) | 0.0 (0.0 to 0.1) | 0.545 | 51.18 (−9.95 to 153.81) up to 02/2015 | 0.118 |
| | | | | −27.10 (−42.97 to −6.81) after 02/2015 | **0.012** |
| **Total Hospital Departments** | | | | | |
| Total CRKP Bacteremia | 0.1 (0.0 to 0.2) | 0.0 (0.0 to 0.1) | 0.445 | −9.40 (−29.69 to 16.75) | 0.445 |
| **Adults Clinic** | | | | | |
| Total CRKP Bacteremia | 0.1 (0.0 to 0.3) | 0.1 (0.0 to 0.2) | 0.614 | 56.27 (−6.69 to 161.69) up to 02/2015 | 0.090 |
| | | | | −27.30 (−42.97 to −7.32) after 02/2015 | **0.010** |
| **Adults Clinic Departments** | | | | | |
| Total CRKP Bacteremia | 0.1 (0.0 to 0.2) | 0.1 (0.0 to 0.1) | 0.495 | −8.27 (−28.40 to 17.52) | 0.495 |
| **Adults ICU** | | | | | |
| Total CRKP Bacteremia | 0.3 (0.1 to 1.2) | 0.5 (0.1 to 1.7) | 0.635 | 137.77 (8.59 to 420.63) up to 01/2015 | **0.030** |
| | | | | −28.44 (−51.84 to 6.33) after 01/2015 | 0.098 |

ICU: intensive care unit; CR: carbapenem-resistant; CRKP: carbapenem-resistant *Klebsiella. Pneumoniae*; EVSP: Estimated Value Start Period; EVEP: Estimated Value End Period; CI: Confidence Interval; *p*-value < 0.05 was bold; All estimates derived from Poisson regression models with robust standard errors, seasonality terms and linear or piecewise linear long-term trend: $\log(N) = \beta_0 + \beta_1 t_- + \beta_2 t_+ + \beta_3 \times \sin(2\pi t/12) + \beta_4 \times \cos(2\pi t/12) + \beta_5 \times \sin(4\pi t/12) + \beta_6 \times \cos(4\pi t/12) + \log(\text{patient-days})$ with N being the number of cases and t being time since study start in months ($t_-$ and $t_+$ piecewise linear time terms; when piecewise linear long-term trend was not required, a single time term was used). % Relative changes/year derived as $[\exp(12 \times \beta_{1,2}) - 1] \times 100\%$.

We investigated the correlation between CRKP and antibiotic consumption (Table 2). The pattern of correlation is repeated between total Hospital Clinics and Departments, and

Adults Clinic and Departments. Not only were antibiotics used to treat CR Gram-negative infections such as carbapenems, colistin, tigecycline, fosfomycin, but also all advanced antibiotics and all antibiotics correlated with increased incidence of CRKP bacteremia. The correlation is inverted if the consumption is reported three months earlier regarding colistin, tigecycline and ceftolozane-tazobactam, which is then correlated with decreased incidence of CRKP bacteremia. In the Adults ICU, the correlation is always positive and statistically significant either for the current or one, two and three months before, for carbapenems ($p$-value = 0.033), aminoglycosides ($p$-value < 0.001), fosfomycin ($p$-value < 0.001), and non-advanced antibiotics ($p$-value = 0.025).

**Table 2.** Incidence of Carbapenem-resistant *Klebsiella Pneumoniae* bacteremia and correlation with consumption of antibiotics, January 2013 to December 2018.

| Antibiotics (DDDs/100 Patient-Days) | per (n) DDD | Month 0 | Month −1 | Month −2 | Month −3 | IRR | 95% CI | *p*-Value |
|---|---|---|---|---|---|---|---|---|
| **CRKP Bacteremia Correlation with Antibiotics** | | | | | | | | |
| **Total Hospital Clinics** | | | | | | | | |
| Carbapenems | 1 | | | ◊ | | 1.54 | (1.05, 2.25) | **0.026** |
| Aminoglycosides | 1 | | ◊ | ◊ | | 2.34 | (1.09, 5.05) | **0.030** |
| Fluoroquinolones | 1 | | | | ◊ | 1.35 | (0.98, 1.86) | 0.067 |
| Colistin | 1 | | | | ◊ | 0.48 | (0.24, 0.96) | **0.037** |
| Tigecycline | 1 | ◊ | | | | 1.48 | (0.96, 2.29) | 0.077 |
| Tigecycline | 1 | | | | ◊ | 0.46 | (0.27, 0.77) | **0.003** |
| Fosfomycin | 0.1 | ◊ | ◊ | | | 1.14 | (1.01, 1.29) | **0.031** |
| Advanced Antibiotics | 1 | | ◊ | | | 1.26 | (1.00, 1.58) | **0.046** |
| All Antibiotics | 10 | | ◊ | ◊ | | 2.55 | (0.96, 6.78) | 0.060 |
| **Total Hospital Departments** | | | | | | | | |
| Monobactams | 0.1 | ◊ | ◊ | | | 0.30 | (0.09, 0.98) | **0.047** |
| Carbapenems | 1 | | | ◊ | | 1.79 | (1.26, 2.55) | **0.001** |
| Colistin | 1 | | | | ◊ | 0.17 | (0.04, 0.74) | **0.019** |
| Tigecycline | 1 | ◊ | | | | 2.68 | (0.92, 7.80) | 0.071 |
| Tigecycline | 1 | | | | ◊ | 0.18 | (0.05, 0.69) | **0.013** |
| Fosfomycin | 0.1 | | | ◊ | | 0.75 | (0.54, 1.03) | 0.078 |
| Ceftolozane—tazobactam | 0.1 | ◊ | | | | 3.92 | (1.06, 14.42) | **0.040** |
| Ceftolozane—tazobactam | 0.1 | | | | ◊ | 0.11 | (0.01, 1.03) | 0.053 |
| **Adults Clinic** | | | | | | | | |
| Monobactams | 0.1 | ◊ | | | | 0.67 | (0.48, 0.94) | **0.021** |
| Carbapenems | 1 | | ◊ | ◊ | | 1.36 | (1.10, 1.67) | **0.004** |
| Aminoglycosides | 1 | | ◊ | ◊ | | 1.95 | (1.17, 3.25) | **0.010** |

**Table 2.** *Cont.*

| Antibiotics (DDDs/100 Patient-Days) | per (n) DDD | Month 0 | Month −1 | Month −2 | Month −3 | IRR | 95% CI | *p*-Value |
|---|---|---|---|---|---|---|---|---|
| **CRKP Bacteremia Correlation with Antibiotics** | | | | | | | | |
| Fluoroquinolones | 1 | | | | ◊ | 1.23 | (1.03, 1.48) | **0.023** |
| Colistin | 1 | | | | ◊ | 0.67 | (0.45, 0.99) | **0.046** |
| Tigecycline | 1 | ◊ | | | | 1.27 | (0.96, 1.67) | 0.091 |
| Tigecycline | 1 | | | | ◊ | 0.63 | (0.46, 0.87) | **0.005** |
| Fosfomycin | 0.1 | ◊ | ◊ | | | 1.08 | (1.00, 1.16) | **0.040** |
| Advanced Antibiotics | 1 | | ◊ | ◊ | | 1.19 | (1.04, 1.36) | **0.012** |
| All Antibiotics | 10 | | ◊ | ◊ | | 2.46 | (1.19, 5.09) | **0.016** |
| **Adults Clinic Departments** | | | | | | | | |
| Monobactams | 0.1 | ◊ | | | | 0.46 | (0.23, 0.93) | **0.031** |
| Carbapenems | 1 | | | ◊ | | 1.29 | (1.01, 1.64) | **0.044** |
| Aminoglycosides | 1 | | | ◊ | | 2.20 | (1.00, 4.85) | 0.051 |
| Colistin | 1 | ◊ | | | | 2.33 | (0.90, 6.04) | 0.082 |
| Colistin | 1 | | | | ◊ | 0.34 | (0.13, 0.86) | **0.023** |
| Tigecycline | 1 | ◊ | | | | 2.08 | (0.97, 4.46) | 0.060 |
| Tigecycline | 1 | | | | ◊ | 0.32 | (0.13, 0.77) | **0.011** |
| Fosfomycin | 0.1 | | | ◊ | | 0.82 | (0.66, 1.02) | 0.074 |
| Ceftolozane—tazobactam | 0.1 | ◊ | | | | 2.31 | (1.02, 5.27) | **0.046** |
| Ceftolozane—tazobactam | 0.1 | | | | ◊ | 0.28 | (0.08, 0.99) | **0.048** |
| Adults ICU | | | | | | | | |
| Carbapenems | 10 | | ◊ | | | 1.26 | (1.02, 1.55) | **0.033** |
| Aminoglycosides | 10 | ◊ | | | | 2.89 | (1.91, 4.38) | **<0.001** |
| Colistin | 10 | | ◊ | | | 1.56 | (0.95, 2.58) | 0.080 |
| Fosfomycin | 10 | ◊ | | | | 2.52 | (1.53, 4.14) | **<0.001** |
| Non Advanced Antibiotics | 10 | | | | ◊ | 1.19 | (1.02, 1.38) | **0.025** |

IRR: incidence rate ratio; CI: Confidence Interval; ICU: intensive care unit; CRKP: carbapenem-resistant *Klebsiella Pneumoniae*; Symbol ◊ denotes whether the association refers to the current month consumption (month 0) value, lagged values (months −1, −2, −3) or averaged values over more than one month. Incidence Rate Ratios (IRR) refers to increases in consumption denoted in column labeled "per (n) DDD". *p*-value < 0.05 was bold. All estimates derived from Poisson regression models with robust standard errors, seasonality effects and spline terms of time: $\log(N) = \beta_0 + \beta_1 V + \beta_2 S_1(t) + \beta_3 S_2(t) + \beta_4 S_3(t) + \beta_5 \times \sin(2\pi t/12) + \beta_6 \times \cos(2\pi t/12) + \beta_7 \times \sin(4\pi t/12) + \beta_8 \times \cos(4\pi t/12) + \log(\text{patient-days})$ with N being the number of cases, t being time since study start in months, S(t) being spline terms of t and V referring to the current month covariate (month 0) value, lagged values (months −1, −2, −3) or averaged values over more than one month. Incidence Rate Ratios (IRR) derived as $[\exp(n \times \beta_1) - 1] \times 100\%$ with n given in column labeled "per (n) DDD".

The correlation results between CRKP bacteremia and infection control interventions are presented in Table 3. Notably, in terms of hand hygiene solutions, particularly in the Adults ICU, each increment in the consumption of scrub disinfectant solutions two months

earlier was correlated with a decreased incidence of CRKP bacteremia (*p*-value = 0.008). Regarding the infection control measure of isolation, in the Adults ICU, every increase in the isolation of patients with MDR pathogens during the current month correlated with an increased incidence of CRKP bacteremia (*p*-value < 0.001). Conversely, two months before, it correlated with a reduced incidence of carbapenem-resistant Gram-negative bacteremia (*p*-value = 0.001 and 0.008, respectively).

**Table 3.** Incidence of Carbapenem-resistant *Klebsiella Pneumoniae* bacteremia and correlation with infection control interventions, January 2013 to December 2018.

| Infection Control Interventions | Per (n) Unit | CRKP Bacteremia Correlation with Infection Control Interventions | | | | | | |
| | | Month 0 | Month −1 | Month −2 | Month −3 | IRR | 95% CI | *p*-Value |
|---|---|---|---|---|---|---|---|---|
| **CRKP Bacteremia** | | | | | | | | |
| **Adults ICU** | | | | | | | | |
| % Isolations /Admissions | 10 | ◊ | | | | 2.60 | (1.54, 4.39) | **<0.001** |
| % Isolations /Admissions | 10 | | | ◊ | | 0.35 | (0.13, 0.97) | **0.044** |
| L of Scrub Disinfectant sol/1000 patient-days | 10 | | | ◊ | | 0.74 | (0.59, 0.93) | **0.008** |

IRR: incidence rate ratio; CI: Confidence Interval; ICU: intensive care unit; CR: carbapenem-resistant; CRKP: carbapenem-resistant *Klebsiella Pneumoniae*; Symbol ◊ denotes whether the association refers to the current month (month 0) value, lagged values (months −1, −2, −3) or averaged values over more than one month. Incidence Rate Ratios (IRR) refers to increases denoted in column labeled "per (n) units". *p*-value < 0.05 was bold. All estimates derived from Poisson regression models with robust standard errors, seasonality effects and spline terms of time: $\log(N) = \beta_0 + \beta_1 V + \beta_2 S_1(t) + \beta_3 S_2(t) + \beta_4 S_3(t) + \beta_5 \times \sin(2\pi t/12) + \beta_6 \times \cos(2\pi t/12) + \beta_7 \times \sin(4\pi t/12) + \beta_8 \times \cos(4\pi t/12) + \log(\text{patient-days})$ with N being the number of cases, t being time since study start in months, S(t) being spline terms of t and V referring to the current month covariate (month 0) value, lagged values (months −1, −2, −3) or averaged values over more than one month. Incidence Rate Ratios derived as $\exp(n \times \beta_1)$ with n given in column labeled "per (n)".

## 4. Discussion

Over the course of this 6-year program, comprising multimodal infection control interventions, our primary objective was to prevent and manage CR Gram-negative bacteremia, with a specific focus on CRKP, in a 300-bed tertiary-care hospital in Athens, Greece. Despite the significant enhancements in the implemented interventions throughout the study, the overall decrease in the incidence of CRKP bacteremia did not achieve statistical significance. Notably, after the second year of the study, a statistically significant negative relative change per year was observed in the context of CRKP bacteremia. The primary explanation for this outcome lies firstly in the duration of the current study and secondly in the high endemic prevalence of CRKP strains within the community and long-term care healthcare facilities in Greece [12–14]. It is conceivable that extending the study period might have been necessary to yield statistically significant overall improvements in the incidence of CRKP bacteremia, particularly in the face of enduring endemicity.

In various studies conducted in countries with a high prevalence of CRKP, a reduction in CRKP incidence within the hospital setting has been reported following the implementation of intensified infection control measures. These measures include enhanced hand hygiene, active surveillance coupled with contact precautions, educational initiatives, and policies involving audits and feedback [20,21]. However, there appears to be a limitation in the efficacy of such measures when the incidence of CRKP has reached endemic levels. The implementation of more stringent measures within the hospital setting may not result in further decreases [25,26]. The optimal scenario would involve avoiding sustained hospital admissions of asymptomatic CRKP carriers [27] and preventing potential unidentified transmissions of CRKP strains among hospitalized patients [28]. When similar infection control programs were implemented across a broad geographic area and in multiple health-

care centers, both the endemic plateau of CRKP and in-hospital dissemination showed significant decreases [29,30].

This phenomenon is largely explained by the population ecology of infectious diseases and the misuse of antibiotics, both in the community and hospital settings [31,32]. In our study, our focus was on providing quantitative data on infection control interventions and outcomes not just for the entire hospital but also for each division. While the incidence of CRKP bacteremia exhibited a decreasing trend in the overall hospital, in the Adults ICU an absolute increase was recorded, although not statistically significant. In the first two years of the study, exogenous high-risk clones isolated from critically ill patients have become endemic in Adults ICU. This was very difficult to be inverted during the last four years of the study and mainly achieved a halt of spread in the rest of the hospital. This observation underscores the necessity for a specifically designed Antimicrobial Stewardship Program (ASP) tailored for critically ill patients, incorporating interventions with a customized frequency, evaluation, and implementation, as demonstrated positively in other studies [33,34]. In our study, not only specific antibiotics used to treat CRKP bacteremia, but also all classes of antibiotics, correlated positively with an increased incidence of these infections, as it is mentioned in other studies [35]. In hospital divisions the correlation was negative when the consumption was reported three months earlier with a decreased incidence of CRKP bacteremia, suggesting that these classes of antibiotics are less likely to be administered for prolonged periods. Conversely, in the Adults ICU, the correlation was consistently positive and statistically significant, whether considering the current month or the preceding three months. This implies that critically ill patients with CRKP bacteremia in the ICU are more likely to be hospitalized for prolonged periods. The latter finding emphasizes the importance of exploring alternative antimicrobial strategies with the aim of developing novel therapeutics against infectious diseases [36]. Until now, various approaches have been explored for their application in antimicrobial therapy with promising results [37].

An additional finding in the present study is that the increased consumption of scrub disinfectant solutions, even two months earlier, correlated with a reduced incidence of CRKP bacteremia, suggesting a lasting post-effect for this intervention. Regarding the isolation of MDR carrier patients during the current month, there was a positive correlation with an increased incidence of CRKP bacteremia in the Adults ICU, implying direct and active surveillance for these pathogens. Conversely, the isolation of MDR carrier patients two months earlier had a negative correlation, with a decreased incidence of CRKP bacteremia, indicating the effectiveness of infection control measures. The latter aspect remains a matter of general concern, especially considering the escalating trends of CR pathogens in healthcare facilities within the context of the COVID-19 pandemic [38,39].

A notable strength of our study is the prospective collection of data over a six-year program. Analyzing findings on a clinic and department level enabled more precise and insightful conclusions. However, there are three limitations worth noting: firstly, this is a single-center study, implying that there may be local factors influencing outcomes that limit generalizability to other centers; secondly, the performance of bacterial typing to seek for nosocomial spread is lacking in order to highlight the late four-year results; thirdly, given our investigation into associations between multiple outcomes and various potential predictors across different hospital sectors, numerous models have been fitted and numerous hypotheses have been tested. Consequently, we present unadjusted *p*-values, which cannot entirely rule out some inflation of the Type I error beyond 0.05 [40].

## 5. Conclusions

This study offers valuable insights into the impact of a six-year infection control program on the incidence of CRKP bacteremia in a private hospital in Greece. The inclusion of multimodal interventions in routine clinical practice, with significant adherence, provides a comprehensive view. While the overall result in the total hospital showed a decreasing trend in the incidence of CRKP bacteremia, it did not reach statistical significance, sug-

gesting the challenge of maintaining an endemic plateau that is difficult to overcome with the applied measures during the study period. Notably, in the Adults ICU, the increased consumption of scrub disinfectant solutions correlated with a reduced incidence of CRKP bacteremia, showcasing a durable post-effect. Additionally, the increased rate of isolation of MDR carriers correlated with a decreased incidence of CRKP bacteremia, emphasizing the effectiveness of infection control interventions in preventing the spread of CRKP strains within the hospital. This effect remains crucial, especially in the context of the global increase in antibiotic use in hospitals and the spread of MDR Gram-negative pathogens during the COVID-19 pandemic. Our findings highlight the need for additional strategies to further contain the incidence of CRKP, considering the evolving landscape of healthcare challenges and infectious diseases.

**Author Contributions:** A.P. (Amalia Papanikolopoulou) and L.V. were involved in the conception of the study; A.P. (Amalia Papanikolopoulou) was involved in data acquisition and wrote the manuscript; D.K. and A.S. contributed to data acquisition and methodology; A.P. (Anastasia Pangalis) and G.C. checked and approved the authenticity of the raw data; N.P. analyzed the data; P.G.-K. reviewed and edited the manuscript; M.K. was involved in the supervision of the study and reviewed and edited the manuscript. All authors have read and agreed to the published version of the manuscript.

**Funding:** This research received no external funding.

**Institutional Review Board Statement:** The study was approved by the Ethics Committees of the National and Kapodistrian University of Athens (approval No: 1718016673—25 January 2018) and Athens Medical Center (approval No: KM140678—26 July 2017).

**Informed Consent Statement:** Patient consent was waived due to the anonymous analysis of clinical data.

**Data Availability Statement:** Data are contained within the article.

**Acknowledgments:** We are thankful to the Infection Control Committee of the hospital. The opinions presented in this article are those of the authors, and do not necessarily represent those of their institutions.

**Conflicts of Interest:** The authors declare no conflict of interest.

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
