# Peer review of "Carbapenem-Resistant Klebsiella pneumoniae Bacteremia: Counterbalance between the Endemic Load and the Infection Control Program in a Hospital"

_2813-9054, doi:10.3390/amh69020009_

Round 1

Reviewer 1 Report

Comments and Suggestions for Authors

Good job.

Comments

-Line 57: Please use a recent reference for CRKP in Greece

Emergence and persistent spread of carbapenemase-producing Klebsiella pneumoniae high-risk clones in Greek hospitals, 2013 to 2022.

Tryfinopoulou K, Linkevicius M, Pappa O, Alm E, Karadimas K, Svartström O, Polemis M, Mellou K, Maragkos A, Brolund A, Fröding I, David S, Vatopoulos A, Palm D, Monnet DL, Zaoutis T, Kohlenberg A; Greek CCRE study group; Members of the Greek CCRE study group.

Euro Surveill. 2023 Nov;28(47):2300571. doi: 10.2807/1560-7917.ES.2023.28.47.2300571.

-Figure 2: Correct the title of graphics

-Table 1: Why don’t you use bold writing for statictically significant values?

-Table 1: Results for Adults ICU, with the statistically significant p values, should be more annotated.

-Line 193: This has been stated before. Please refer to

Antimicrob Resist Infect Control . 2019 Feb 14:8:38. doi: 10.1186/s13756-019-0484-8.

Influence of antibiotic pressure on multi-drug resistant Klebsiella pneumoniae colonisation in critically ill patients

Jesus Ruiz 1, Monica Gordon 1, Esther Villarreal 1, Juan Frasquet 2, María Ángeles Sánchez 3, María Martín 1, Álvaro Castellanos 3, Paula Ramirez 3

-Did you look for the exact gene providing K.pneumoniae isolates carbapenem resistance?

-Line 266: saw?

-Line 291: Please add to limitations the lack of bacterial typing to seek for nosocomial spread.

-This study offers valuable insights into the impact of a six-year infection control program on the incidence of CRKP bacteremia in a private hospital in Greece.

Reviewer 2 Report

Comments and Suggestions for Authors

Interesting study evaluating the relationship between different infection control and antimicrobial stewardship measures, and carbapenem-resistant gram-negative bacteremia, focusing on carbapenem-resistant K. pneumoniae.

Study design is solid and presents data over a 6-year period. Manuscript is well written and descriptive. There are some minor points in the methods that need more description. Also some issues in the discussion that need clarifications.

Overall, a well performed study with important and practical findings, particularly for high-prevalence settings.

Some minor comments:

Figure 1 doesn’t seem necessary, as it’s covered by the in-text description.

Methods

Line 95: can the authors mention which antimicrobials were restricted and preauthorized?

Line 98: which patients were screened for carriage? Can authors briefly describe the isolation precautions applied to MDR carriers?

Figure 2 please correct the title of 2(a)

In the limitations, I’d suggest adding that some conclusions were assumed based on findings and causal relationships were not established through study design (eg.durable post-effect of hand hygiene scrub use, correlation between MDR carrier incidence and reduced CRKP).

Reviewer 3 Report

Comments and Suggestions for Authors

Dear Authors,

Your article entitled "Carbapenem-resistant Klebsiella pneumoniae bacteremia: counterbalance between the endemic load and the infection control program in a hospital" has been reviewed,

This article deserves attention since it highlight on a very important topic, related to public health. In this work you tried to understand a counterbalance between the endemic load and the infection prevention and control program regarding one of the major resistant strains "Carbapenem-resistant Klebsiella pneumoniae (CRKP)" in one tertiary healthcare center in Athens, Greece.

The paper must be improved in its Quality of Presentation, its design and English Language.

Kindly find below a list of my comments, minor and major ones:

Minor Comments:

01- In the Whole manuscript, You are kindly invited to put "P-value" instead of "p-value". And "P" instead of "p"

02- In the Whole manuscript, You are kindly asked to put the bacterial name in italic. Example: "Klebsiella pneumoniae" in the keywords list.

03- In the Whole manuscript, the first time you use the name of a bacterial strain you are invited to put its full name in italic followed by its abbreviation between parenthesis. Example: Pseudomonas aeruginosa (P. aeruginosa). Then you can use the abbreviation instead of its full name.

04- As a General Point, Regarding the Introduction, I suggest to start with a small paragraph in which you introduce Klebsiella pneumoniae, then you can talk about its antimicrobial resistance, and then you talk about CRKP. I suggest this review as reference for this point:

Reference 01: General Overview of Klebsiella pneumonia: Epidemiology and the Role of Siderophores in Its Pathogenicity

05- In the Introduction section, Line 48, You are kindly invited to replace "HAI" by "HAIs", since it is infections.

06- In the Introduction section, Lines 47-49, After (CRE), You are kindly invited to put a reference for this idea.

07- In the Introduction section, Lines 55-56, You are kindly invited to explain for readers what is VIM, KPC, NDM and OXA-48-Like.

08- In the Introduction section, Lines 78-79, You are kindly invited to replace "carbapenem-resistant Klebsiella pneumoniae (CRKP)" by "CRKP". 

09- Concerning Figure 1, The words in this figure are very small, you are kindly invited to change the size of words.

10- In the Line 157, Authors are invited to correct the reference form for (15).

11- Concerning Figure 2, The title (in the figure) of the section (a) is not correct, you are kindly asked to replace "Carbapenem-resistant Klebsiella pneumoniae" by "Gram-negative".

12- Concerning Table 2, You are kindly invited to put all the numbers "10, 1 and 0.1) on the same line (Aligned). 

13- Concerning the sample size, you are kindly asked to calculate the power of this study.

14- Since it is a paper tackling the danger of one of the well-known resistant strain which is (CRKP), Authors are invited to talk about the importance of fighting these strains and about the importance of finding new antibacterial treatment to fight Antimicrobial Resistance (AMR). I suggest the following papers as references (you can use these papers or related ones) for this point:

Reference 02: Strategies to Overcome Antimicrobial Resistance (AMR) Making Use of Non-Essential Target Inhibitors: A Review

Reference 03: Towards new antibiotics classes targeting bacterial metallophores

Reference 04: Where are all the new antibiotics? The new antibiotic paradox

Major Comments:

01- One of my major comments, is related to the period of this study, these information were collected in the period between 2013 and 2018, so it is somehow out of date. When we talk about major MDR strains we are talking about a yearly change (Mostly increase rate) in the prevalence of these strains. In addition, these data were collected before the Pandemic and as you know the COVID-19 has a great impact on public health and mainly on infectious diseases trends, resistance, and Infection Prevention and Control. So mostly this is the biggest limitation in this study.

02- In the Results section, Lines 192-195, When you talked about about the effect of the consumption of four different antibiotics earlier in 3 months. This information doesn't match with results shown in the Table 2 for Fosfomycin, There no evidence about Fosfomycin taken 3 months earlier. Can you please explain this point? 

03- Table 3 is fully of empty cells!!! Is it normal? Kindly explain this point.

Best Regards,

Comments on the Quality of English Language

Dear Authors,

Your present manuscript needs several English Language corrections and paraphrasing.

Reviewer 4 Report

Comments and Suggestions for Authors

The authors investigated the impact of antibiotic use and infection control measures on CRKP bacteremia in a tertiary-care hospital from 2013 to 2018. Findings show a non-significant decrease in CRKP bacteremia incidence. Increased antibiotic use was associated with higher CRKP bacteremia rates, while enhanced disinfection and isolation practices, especially in the Adult ICU, were linked to lower incidence rates. The study is important and suggests that comprehensive infection control strategies can decrease CRKP spread. There are several comments as followings:

1.     Why authors only included carbapenem-resistant K. pneumoniae, P. aeruginosa, and A. baumanii? Why Enterobacter spp. was excluded?

2.     In terms of correlation between antibiotics use and incidence of CRKP bacteremia, authors should clear in the abstract that which one is positive collected and which one is negative correlated. As we know the incidence of CRKP bacteremia will increase the antibiotics use and inappropriate use of antibiotics will also increase the incidence of CRKP bacteremia. Please discuss in your manuscript which one is the major factor.

3.     In Figure 2, the captions of a and b are the same. Should be corrected.

4.     Abbreviations throughout the text need to be standardized. The carbapenem-resistant Klebsiella pneumoniae (CRKP) appeared in line 43, 54 78. The authors should check all throughout the manuscript.

Comments on the Quality of English Language

The English is fine.

Round 2

Reviewer 3 Report

Comments and Suggestions for Authors

Dear Authors,

The revised version of your manuscript have been reviewed,

The article is better in its present form, its design, English language and Style of Results' presentation is better than the previous one,

I just have to comments:

The first one is a minor one related to the importance of mentioning the diseases caused by the pathogenic agent "Klebsiella pneumoniae", such as pneumonia, urinary tract infection, bloddstream infection, wound or surgical site infection and meningitis.

You are invited to use the following articles as references for this point:

Reference 01: Invasive infection caused by Klebsiella pneumoniae is a disease affecting patients with high comorbidity and associated with high long-term mortality

Reference 02: Antimicrobial Susceptibilities and Laboratory Profiles of Escherichia coliKlebsiella pneumoniae, and Proteus mirabilis Isolates as Agents of Urinary Tract Infection in Lebanon: Paving the Way for Better Diagnostics

Reference 03: Klebsiella pneumoniae Isolates from Meningitis: Epidemiology, Virulence and Antibiotic Resistance

Reference 04: Prevalence and characteristics of surgical site hypervirulent Klebsiella pneumoniae isolates.

The second one is a major one, concerning the table 3, why first cells are free? Please inform me birefly.

Best Regards,

Author Response

Manuscript Ref.: No. amh-2922908
Title: Carbapenem-resistant Klebsiella pneumoniae bacteremia: counterbalance between the endemic load and the infection control program in a hospital
acta microbiologica hellenica

Answer to Reviewers' comments #3 round2

The revised version of your manuscript has been reviewed,

The article is better in its present form, its design, English language and Style of Results' presentation is better than the previous one,

Minor Comments:

Comment 1

I just have to comments:

The first one is a minor one related to the importance of mentioning the diseases caused by the pathogenic agent "Klebsiella pneumoniae", such as pneumonia, urinary tract infection, bloddstream infection, wound or surgical site infection and meningitis.

You are invited to use the following articles as references for this point:

Reference 01: Invasive infection caused by Klebsiella pneumoniae is a disease affecting patients with high comorbidity and associated with high long-term mortality

Reference 02: Antimicrobial Susceptibilities and Laboratory Profiles of Escherichia coli, Klebsiella pneumoniae, and Proteus mirabilis Isolates as Agents of Urinary Tract Infection in Lebanon: Paving the Way for Better Diagnostics

Reference 03: Klebsiella pneumoniae Isolates from Meningitis: Epidemiology, Virulence and Antibiotic Resistance

Reference 04: Prevalence and characteristics of surgical site hypervirulent Klebsiella pneumoniae isolates.

Answer to Comment 1

We thank the reviewer for his/her valuable comments to improve the article. We have added in the introduction, in the paragraph introducing Klebsiella pneumoniae, the above references (lines 44-51, ref 2-5).

Major Comments:

 Comment 1

The second one is a major one, concerning the table 3, why first cells are free? Please inform me briefly.

Answer to Comment 1

We thank the reviewer for this comment. In table 3 we present CRKP Bacteremia correlation with infection control interventions. We have omitted from the table the lines regarding Total Hospital Clinics / Total Hospital Departments / Adults Clinic / Adults Clinic Departments because the correlation results were not significant and seem empty.